# Current Methods to Unravel the Functional Properties of Lysosomal Ion Channels and Transporters

**DOI:** 10.3390/cells11060921

**Published:** 2022-03-08

**Authors:** Margherita Festa, Velia Minicozzi, Anna Boccaccio, Laura Lagostena, Antonella Gradogna, Tianwen Qi, Alex Costa, Nina Larisch, Shin Hamamoto, Emanuela Pedrazzini, Stefan Milenkovic, Joachim Scholz-Starke, Matteo Ceccarelli, Alessandro Vitale, Petra Dietrich, Nobuyuki Uozumi, Franco Gambale, Armando Carpaneto

**Affiliations:** 1Department of Biology, University of Padova, Via Ugo Bassi 58/B, 35131 Padova, Italy; margherita.festa@unipd.it; 2INFN, Department of Physics, University of Rome Tor Vergata, Via della Ricerca Scientifica 1, 00133 Rome, Italy; velia.minicozzi@roma2.infn.it; 3Institute of Biophysics, National Research Council, Via De Marini 6, 16149 Genoa, Italy; anna.boccaccio@ibf.cnr.it (A.B.); laura.lagostena@ibf.cnr.it (L.L.); antonella.gradogna@ibf.cnr.it (A.G.); joachim.scholzstarke@ibf.cnr.it (J.S.-S.); franco.gambale@gmail.com (F.G.); 4Department of Earth, Environment and Life Sciences (DISTAV), University of Genoa, Viale Benedetto XV 5, 16132 Genoa, Italy; qitianwen318@126.com; 5Department of Biosciences, University of Milan, 20133 Milan, Italy; alex.costa@unimi.it; 6Department of Biology, Friedrich-Alexander-Universität Erlangen-Nürnberg (FAU), 91058 Erlangen, Germany; ninalarisch@gmx.de (N.L.); petra.dietrich@fau.de (P.D.); 7Collaborative Research Institute for Innovative Microbiology, The University of Tokyo, 1-1-1 Yayoi, Bunkyo-ku, Tokyo 113-8657, Japan; uhamamoto@g.ecc.u-tokyo.ac.jp; 8Institute of Agricultural Biology and Biotechnology, National Research Council, Via Bassini 15, 20133 Milan, Italy; emanuela.pedrazzini@ibba.cnr.it (E.P.); alessandro.vitale@ibba.cnr.it (A.V.); 9Department of Physics, University of Cagliari, 09042 Monserrato, Italy; milenkovic.st@gmail.com (S.M.); matteo.ceccarelli@dsf.unica.it (M.C.); 10IOM-CNR Unità di Cagliari, Cittadella Universitaria, 09042 Monserrato, Italy; 11Department of Biomolecular Engineering, Graduate School of Engineering, Tohoku University, Sendai 980-8579, Japan; uozumi@tohoku.ac.jp

**Keywords:** lysosomes, ion channels, transporters, plant vacuole, patch-clamp

## Abstract

A distinct set of channels and transporters regulates the ion fluxes across the lysosomal membrane. Malfunctioning of these transport proteins and the resulting ionic imbalance is involved in various human diseases, such as lysosomal storage disorders, cancer, as well as metabolic and neurodegenerative diseases. As a consequence, these proteins have stimulated strong interest for their suitability as possible drug targets. A detailed functional characterization of many lysosomal channels and transporters is lacking, mainly due to technical difficulties in applying the standard patch-clamp technique to these small intracellular compartments. In this review, we focus on current methods used to unravel the functional properties of lysosomal ion channels and transporters, stressing their advantages and disadvantages and evaluating their fields of applicability.

## 1. Introduction

Lysosomes are acidic organelles, pH of about 4.6 [1], considered as the digestive system of the animal cell. They act as the major compartment for detoxification of both the outer and the inner content of the cell. In fact, lysosomes represent the key players in degradation, recycling, autophagy, cell death, cell proliferation, cell defence, immunity–autoimmunity processes and therefore in maintenance of cellular homeostasis [2,3,4,5].

Of prime importance to conduct these diverse cellular functions is a highly organized ion homeostasis and control of the organellar pH, which is achieved via a set of ion channels and transporters [6,7]. Using proteomics approaches on purified organellar membranes, new lysosomal membrane (LM) proteins have been discovered [8,9], highlighting new features and functions of these organelles.

Lysosomes are engaged in cross-talk with each other, with other organelles like mitochondria [10] and with proteins/receptors outside the organelle, highlighting the existence of complex cellular mechanisms of regulation.

Supporting their vital role, the correct function of these organelles is disturbed in a group of human pathologies known as lysosomal storage disorders, LSDs [11]. More than 60 human LSDs have been identified, which can be divided into two subgroups [12]: (i) LSDs caused by the deficiency of specific luminal enzymes; and (ii) LSDs caused by defective lysosomal transmembrane proteins (LTPs) essential for the transport of solutes. Therefore, current research on LTPs provides the basis for a better understanding of LSDs and future therapeutic intervention. Besides their involvement in protein degradation and storage functions, lysosome channels are also potentially essential for viral infections in humans [13,14].

### 1.1. Main Families of Lysosomal Channels and Transporters

The lysosomal ion channels and transporters known so far belong to a limited number of protein families (Table 1).

Members of the CLC (chloride channels and transporters) family, specifically CLC-6 and CLC-7, serve to transport Cl^−^ across the LM. CLC-7 also contributes to the efficient acidification of the lysosome [15] and has been implied in Alzheimer’s disease due to an impairment of the amyloid fibril clearance within lysosomes [16]. For a review on the CLC family, see [17].

The SLC38 (solute carrier 38) family of sodium-amino acid co-transporters [18,19] is involved in the maintenance of amino acid levels inside the lysosomal lumen, which is crucial for the regulation of cell growth and catabolism [20,21]. The SLC38A7 transporter has been proposed as a new target for glutamine-related anti-cancer drugs, as it is required for cancer cell growth [22].

NHE Na^+^/H^+^ exchangers, in particular NHE3, NHE5 and NHE6 [23] are involved in Na^+^ accumulation within the lysosomal lumen.

TPCs (Two Pore Channels) are cation channels involved in Ca^2+^ signalling and activated by the ligands NAADP and PI(3,5)P_2_ [24,25,26,27,28]. TPC proteins may be possible drug targets for the treatment of Ebola and SARS-CoV-2 viruses infection, since they control virus entry into the host cell [13,14,29,30]. Moreover, TPCs have been shown to be involved in neurodegenerative Parkinson’s disease [31] and in processes of neoangiogenesis [32,33,34].

VGCCs (Voltage-Gated Calcium Channels) regulate the lysosomal fusion with endosomes and autophagosomes and are also required for synaptic vesicle fusion with the plasma membrane and neurotransmitter release [35,36].

The family of TRP (Transient Receptor Potential) channels has been reviewed in [37,38]. The TRPML isoforms 1, 2 and 3 are present at the endolysosomal membrane and form non-selective cation channels permeable to various cations (Na^+^, Ca^2+^, Fe^2+^, Zn^2+^) and activated by PI(3,5)P_2_ [39]. TRPML1 mutations are involved in mucolipidosis type IV [40], an LSD presenting an impaired neurodevelopment. TRPML2 seems to be involved in the regulation of the immune response [41]. The varitint-waddler (Va) deafness mutation in TRPML3 is connected to cell degeneration [42,43,44,45].

P2X4 is a Ca^2+^-permeable channel in the lysosomal membrane. It was recently shown that calcium release through P2X4 activates calmodulin to promote endolysosomal membrane fusion [46].

BK (calcium-activated big conductance K^+^ channel) has long been thought to be the only K^+^ channel on the lysosome, but a new transmembrane protein, named TMEM175, has been found to be a lysosomal potassium-selective channel [47]. BK channels as well as TMEM175 may alleviate LSDs by providing positive feedback regulation of lysosomal Ca^2+^ release [48]. Hence, upregulating BK may be a potential therapeutic strategy. Moreover the TMEM175 K^+^ channel is supposed to be important for maintaining the membrane potential and pH stability in lysosomes [47,49] and may play a role in the pathogenesis of Parkinson’s disease [50].

Leucine-rich repeat containing family 8 (LRRC8) proteins, which form volume-regulated Cl^−^/anion channels (VRACs), have been shown to localize not exclusively on plasma membrane, but also on lysosome membranes and generate large anion currents in response to low cytoplasmic ionic strength conditions [51].

V-type (vacuolar type) H^+^-ATPase is the proton pump that acidifies the lysosomal lumen by transferring two protons into the lysosome for each consumed ATP molecule [52], providing the proton-motive force necessary for the function of H^+^-dependent exchangers.

CLN7 (Ceroid Lipofuscinosis Neuronal 7) was very recently identified as a novel endolysosomal chloride channel [53]. It mediates a lysosomal chloride conductance exhibiting properties common to chloride channels. It also promotes the release of lysosomal Ca^2+^ through TRPML1.

Lysosomal ion channels and transporters maintain the lysosome in the conditions necessary for cell survival. Impairment in lysosomal function due to channel and transporter malfunctioning is involved in LSDs, neurodegenerative diseases and cancer [54]. At the same time, overexpression or underexpression of lysosomal proteins may have a therapeutic function. Therefore, it is of utmost importance to be able to investigate lysosomal channels and transporters, being also possible drug targets. However, the difficult manipulation of these proteins due to their intracellular localization in endo-lysosomal submicrometric compartments represents a limiting step in such studies.

### 1.2. Summary of the Experimental Methods to Investigate the Functional Properties of Lysosomal Ion Channels and Transporters

In this review, we present an overview on the techniques available for the functional characterization of lysosomal channels and transporters. A summary of the methods, their respective advantages and disadvantages, together with the lysosomal channels/transporters to which they have been applied is presented in Table 2. References are inserted in the main text, where the different techniques are described in detail.

## 2. Approaches Using Purified Proteins or Native Endolysosomal Membranes

### 2.1. Incorporation into Artificial Membranes or Liposomes

A planar bilayer is an artificial membrane formed across a small hole placed in a thin plastic partition that separates two aqueous compartments (Figure 1). In turn, lipid bilayers may be used to characterize the activity of lysosomal membrane proteins at the level of single molecular events by incorporating either liposomes containing purified proteins (Figure 2) or native membranes vesicles derived from specific compartments/organelles [55].

Lipid bilayers have a very low level of background current noise so that it is possible to record single-channel currents. Accessibility of both sides of the bilayer and the possibility to clamp the membrane voltage allow studies of channel gating, ion conduction and selectivity, effect of ligands, etc.

An advantage of the bilayer approach is that it enables to examine the effect of the lipid environment on the channel, as bilayers may be formed by different types of lipids. Disadvantages include the requirement of a sufficient amount of protein, the fact that the channel is removed from its native environment and that there is no control of protein orientation within the membrane when purified protein is used. However, the main drawback, especially when native membrane vesicles are used, is the presence of impurities, whose activity may be erroneously attributed to the protein of interest.

Single channel events of immunopurified hTPC2 and hTPC1 reconstituted in artificial membranes were observed respectively by [62,63,64].

TRPML1 function was investigated in lipid bilayers using reconstitution of both endosomal vesicles derived from cells over-expressing TRPML1 and liposomes previously dialyzed with TRPML1 protein [65]. TRPML1 reconstituted in lipid bilayers showed spontaneous cation channel activity in the presence of asymmetric K^+^, voltage-dependent activation and multiple sub-conductance states.

### 2.2. Solid-Supported Membrane-Based Electrophysiology

A number of different electrogenic proteins (ion pumps, transporters and channels) have been tested using solid-supported membrane (SSM) electrophysiology [66,67,68,69]. More recently, this technique has been applied to intracellular transporters to overcome the inaccessibilty of endomembranes [70,71].

For such electrical measurements, the SSM is built on a planar gold electrode by depositing an alkane-thiol layer followed by a phospholipid monolayer on top of it (Figure 3). Proteoliposomes containing purified proteins or membrane vesicles and membrane fragments from native tissue containing the protein of interest are adsorbed on this SSM to form a capacitively coupled membrane system. The SSM behaves like a capacitor with the solution in the fluid compartment, and with the gold electrode being the capacitor plates.

Charge translocation, due to the protein’s transport activity, is initiated by providing a substrate or a ligand via rapid solution exchange. The transport-dependent transient currents correspond to the charging of the gold electrode sensor and the charging kinetics depends on the transport activity of the assayed protein. In general, the peak current value is used to analyse the stationary protein transport activity. Membrane fractions (from native tissue or transfected cells) enriched in plasma membrane or specific intracellular membranes can be obtained by sugar gradient fractionation, giving the possibility to study proteins expressed in intracellular organelles in their native environment.

In SSM-based electrophysiology the use of membrane fragments from native tissue or from transfected cells should assure an electrophysiological characterization in the protein’s native environment.

It is possible to investigate the functional properties even of slow transporters, whose transport-related currents are too small for classical electrophysiological experiments. This technique is attractive in the view of establishing screening assays. The major drawback of this technique is that it cannot be used to apply an electrical voltage. Transporter characterization is therefore restricted to transport modes which do not rely on a membrane potential.

SSM-based electrophysiology has been used by Obrdlik and colleagues [67] to study V-ATPase in synaptic vesicles and Na^+^/K^+^-ATPase present in plasma membrane fractions prepared from the rat brain.

It has also been used to characterize rat CLC-7 function and the effect of the disease-causing G213R mutation, responsible for autosomal-dominant osteopetrosis type II [68]. In this study, rCLC-7 and G213R CLC-7 (which is the analogue of human G215R CLC-7) were expressed in CHO cells, alone or together with the accessory subunit OSTM1. Since CLC-7 and G213R CLC-7 did not localize in the plasma membrane, rather respectively in lysosomes and in lysosomes and ER, they used fractions of these intracellular membranes. Lysosomes and ER membranes were enriched in different membrane fractions, as confirmed both by fluorimetric investigation of the different membrane fractions containing fluorescently tagged CLC-7 and by densitometric analysis of CLC-7 Western blots. The membrane fraction was adsorbed to the SSM sensor and currents were generated by fast application of 30 mM NaCl at different pH values. They confirmed that CLC-7 functions as a Cl^−^/H^+^-exchanger, and screened a number of potential chloride channel inhibitors (DIDS and NS5818 inhibited the currents with relatively high affinity). They concluded that mislocalization from lysosomes to ER rather than impaired functionality of G215R CLC-7 is the primary cause of the disease.

### 2.3. Flux Measurements on Purified Lysosomes

Concentrative isotope uptake was previously used for measuring ion fluxes through ion channels in membrane vesicles [72]. Vesicle suspensions were incubated with the ^22^NaCl isotope and the amount of ^22^Na trapped within the vesicles was measured. This procedure allowed to measure a specific ^22^Na uptake and to identify the fraction of vesicles containing Na^+^ channels among a heterogeneous vesicle population [72] Concentrative uptake of ^36^Cl^−^ due to a Cl^−^ gradient was used to determine the conductance properties of Cl^−^ channels extracted from Torpedo plasma membrane and reconstituted into liposomes [73]. The concentrative uptake method was employed to show that the Cl^−^/H^+^ antiporter CLC-7 is a major chloride permeation pathway in lysosomes [15]. Lysosomes isolated from rat liver by differential sedimentation were loaded with high concentrations of unlabelled chloride and then diluted into a buffer containing ^36^Cl^−^. The rapid uptake of ^36^Cl^−^, which was abolished by the external addition of valinomycin, suggested the presence of a specific electrogenic transport pathway for chloride. Additional experiments varying internal anions and in the presence of a pH gradient established the apparent permeability sequence and showed the coupling between Cl^−^ and proton transport. Measurements performed with a Cl^−^ gradient and monitoring the internal pH with the fluorescent dye 2′,7′-bis-(2-carboxyethyl)-5(6)-carboxyfluorescein (BCECF) showed that protons could move against the pH gradient as expected for a Cl^−^/H^+^ antiporter. Finally, the equilibrium potential for H^+^ flux, monitored by BCECF, was measured at a series of theoretical voltages set with K^+^/valinomycin. As a result of these experiments performed on isolated lysosomes, by using concentrative ^36^Cl^−^ uptake combined with fluorescence measurements of proton fluxes, the authors could establish that the lysosomal transport of Cl^−^ and H^+^ is mediated by a Cl^−^/H^+^ antiporter, identified as CLC-7 [15]. This method allows to estimate fluxes in a large number of lysosomes under varying external conditions and maintaining the proteins in their native membrane. It can identify ion transport mechanisms across the lysosomal membrane, however its use to characterize in detail the functional activity of lysosomal membrane transporters seems difficult. It requires performing radioactivity measurements and it does not allow direct and precise control of the membrane potential preventing the study of the voltage dependence of the ion transport.

### 2.4. Patch-Clamp Electrophysiology on Enlarged Lysosomes

Acidic lysosomal compartments in animal cells are small in size (diameter < 500 nm), and this property has strongly limited their use for patch-clamp studies and thus our knowledge about the transporter composition of the lysosome membrane. The importance of patching endolysosomal membranes relates to the fact that, in animal cells, a series of channels have been reported to be localized on both lysosomal and/or endosomal membranes and have been predicted to play roles in signalling events and endomembrane fusion [17,47,74,75,76,77].

A detailed protocol reporting the use of enlarged lysosomes for patch-clamp was published in 2017 [78], but the first patch-clamp recording on endosomal membranes was made possible thanks to the expression of a hydrolysis-deficient SKD1/VPS4B (E235Q) protein in HEK293 cells, which induced the formation of enlarged endosomes (3–6 µm in diameter) by blocking their transition to lysosomes, hence making them accessible to the patch-clamp approach [79]. This strategy allowed the characterization of an endosomal Ca^2+^ channel whose activity was affected by the luminal Cl^−^ concentration [79]. Following this first report, the study of endolysosomal channels showed a strong acceleration thanks to the use of vacuolin-1, a lipid-soluble polycyclic triazine [80]. Vacuolin-1 treatment can selectively increase the size of endosomes and lysosomes from less than 0.5 µm up to 5 µm [75], hence making this compartment accessible to electrophysiological recordings after the mechanical rupture of the plasma membrane [81], Figure 4. Unfortunately, due to the required cell manipulation and mechanical isolation of the endolysosomes, the patch-clamp recordings could be carried out only within a limited time window, from 1 to 3 h after isolation, with a gradually decreasing probability to form high-resistance seals with time [81]. Nevertheless, the enlarged endolysosomes obtained by vacuolin-1 treatment offer a solid basis for the electrophysiological characterization of a series of endomembrane channels [25,74,76,77,82,83].

Vacuolin-1 treatment was first employed for the electrophysiological characterization of TRPML1 (mucolipin 1). In 2008, Dong and colleagues [74] transiently transfected HEK293T cells with an mCherry-TRPML1 construct and performed patch-clamp recordings on native membranes of enlarged endolysosomes in three distinct patch-clamp configurations: Lysosome attached, lysosome luminal-side-out and whole lysosome. These experiments reported the existence of significant inwardly rectifying currents mediated by the permeation of Fe^2+^ [74]. The coupling of a glass chip-based method (planar patch) with the preparation of enlarged endolysosomes led Schieder and colleagues [82] to electrophysiologically characterize another organellar membrane channel: Human TPC2. Characterization of hTPC2 and hTPC1 was also performed by directly patching enlarged endolysosomes. Wang and colleagues [25] proposed that TPC2 was activated by PI(3,5)P_2_ and that the major cation fluxing through its pore was Na^+^. Similarly, also TPC1 was found to transport mainly Na^+^ [77], supporting the view that TPC channels are PI(3,5)P_2_-activated Na^+^-selective channels [25,77]. Nevertheless, other groups succeeded in the study of TPC channels in enlarged endolysosomes and reported that TPC2 were also activated by NAADP [84] and inhibited by cytosolic and luminal magnesium [85]. 

## 3. Approaches Based on Alternative Targeting and Heterologous Expression

### 3.1. Targeting to the Plasma Membrane upon Manipulation of Sorting Signals

Most organelles are not easily amenable to classical uptake experiments or electrophysiological recordings, hampering the analysis of intracellular ion and solute transport processes. One exception to this is the large central vacuole of plant and yeast cells, which is directly and easily accessible after rupture of the plasma membrane. One possibility to circumvent the problem of membrane accessibility is to manipulate the subcellular localization of intracellular transport proteins by altering their targeting route, thus redirecting them to the vacuolar (see Section 3.3) or plasma membranes.

In the secretory or endocytic pathway, coordinated vesicle trafficking delivers transport proteins to the correct destination membrane. Transport vesicles recruit their transmembrane protein cargo upon interaction of adaptor proteins with sorting or internalization signals within the cytoplasmic regions of the cargo proteins. Sorting or internalization signals consist of short, linear motifs, post-translational modifications or three-dimensional structural motifs [86,87,88]. Among the linear sequence motifs, dileucine-based motifs [DE]xxxL[LI] and tyrosine-based motifs YxxØ (where Ø represents a bulky, hydrophobic amino acid) are common. They are recognized by clathrin adaptor complexes (AP1 to AP5), Golgi-localized, γ-ear containing, Arf-binding proteins (GGAs) and stonins 1 and 2. GGAs select the dileucine-motif variant DxxLL and specifically control vesicle sorting at the trans-Golgi network [89], Figure 5. The only identified cargo for stonins is up to now synaptotagmin-1, where stonin 2 directly binds a cluster of basic residues [90]. While stonins have to date only be identified in meazoans and GGAs in metazoans and fungi [91], in general the sorting motifs and trafficking mechanisms are quite well conserved among different phylogenetic groups, including *mammalia*, plants and yeast. Such conservation of trafficking mechanisms can be exploited to manipulate the targeting of intracellular transport proteins for surface expression and functional characterization.

This approach was used in 2000 as a tool for Glucose Transporter 8, GLUT8, formerly GLUTX1; [92]. Ibberson and colleagues cloned this transporter from rat cDNA but were not able to detect glucose uptake in GLUT8-expressing *Xenopus* oocytes. They suspected an internalization signal to prevent sufficient surface expression and therefore mutated a dileucine motif at the N-terminus of GLUT8. This mutation was sufficient to redirect the protein to the plasma membrane and established GLUT8 as a glucose transporter. In a later study, Ref. [93] showed that the canonical dileucine motif ExxxLL is responsible for localization of the native GLUT8 in late endosomes and lysosomes. When a homologous transporter of GLUT8 from *Arabidopsis thaliana* termed ESL1 (ERD Six-Like 1) was investigated, uptake experiments with a mutated N-terminal dileucine-motif revealed its transport capacities for glucose and some other monosaccharides [94]. Confocal microscopy of GFP fusion proteins confirmed that manipulation of the sorting motif redirected ESL1 from the vacuolar to the plasma membrane of tobacco BY2 cells.

Protein redirection to the plasma membrane upon manipulation of sorting/internalization motifs turned out as an elegant approach for functional studies of various lysosomal transporters and ion channels from animal as well as plant cells (Table 3). Among these, human TPC2, which is mainly localized in late endosomes and lysosomes, was studied in the plasma membrane of mammalian cells following disruption of its N-terminal dileucine motif (ESEPLL) via deletion or mutation [95]. Patch-clamp recordings and Ca^2+^ imaging suggested TPC2 to encode an NAADP-gated Ca^2+^-permeable channel [95], a conclusion that has been challenged by other approaches [25], including the expression of the channel in plant vacuoles lacking its endogeneous TPC variant [26,96]; see also Section 3.3. Similar to the mammalian channels, Arabidopsis TPC1 harbors a functional N-terminal dileucine-based motif (EDPLI) with a shorter distance between the acidic residue and the dileucine-motif and was efficiently targeted to the plasma membrane upon mutation or deletion of this sorting signal [97]. Mutation of the sorting signal and subsequent surface expression in HEK293 cells was furthermore used to resolve the crystal structure of Arabidopsis TPC1 [98,99], and for cryo-electron microscopy analysis of the structure of human TPC2 [100].

Two dileucine motifs in the N-terminus of the Cl^−^/H^+^-exchanger CLC-7 were identified as lysosomal sorting motifs in 2010 [101]. Leisle and collaborators [102] generated a dileucine double mutant and analyzed the biophysical properties of the corresponding plasma membrane-residing proteins in *Xenopus* oocytes and in HeLa cells. Both electrophysiological approaches demonstrated CLC-7 to encode a 2Cl^−^/1H^+^-exchanger with strongly outward rectifying properties.

For the transporters and channels mentioned above, the sorting motifs could be clearly identified and specifically modified. However, the targeting information within the membrane protein may not necessarily be encoded by canonical di-leucine or tyrosine-based sorting signals and may thus not be easily predicted. Deletion of the tyrosine-based sorting motif within the C-terminal tail of the lysosomal proton-coupled cystine transporter cystinosin resulted only in a partial redirection to the plasma membrane [103]. This was sufficient to study the proton-translocation for this unusual seven-transmembrane domain protein, which is distantly related to the proton pump bacteriorhodopsin [104,105]. A complete plasma membrane relocalization required the additional deletion of a novel, non-canonical lysosomal sorting motif (YFPQA) in a cytosolic loop [103], which in turn abrogated the transporter function [97]. Different from bacteriorhodopsin, cytinosin is characterized by two PQ-loops, characteristic for all members of this sevenhelical protein family. For another member of this family, PQLC2, the cationic amino acid transport mechanism could be revealed by electrophysiological studies of Xenopus plasma membranes after mutating its C-terminal dileucine-type sorting motif [106,107].

As sorting/internalization motifs are often situated near the N- or C-terminal end of the membrane proteins, their trafficking route or efficiency might be affected by the fusion of protein tags, due to masking effects, as was shown for the late endosomal chloride transporter CLC-6. Contrary to its family member CLC-7, the sorting motif(s) for CLC-6 could not yet be identified [101], but fusion of GFP to its N-terminus caused the partial mislocalization to the plasma membrane in Xenopus oocytes and mammalian cells, where it mediates Cl^−^/H^+^ antiport [108,109]. Another example for the use of a masked sorting or retention signal for functional characterization is Aquaporin-6 (AQP6), which was relocalized from acidic intracellular vesicles to the plasma membrane in mammalian cells upon N-terminal fusion with a GFP- or HA-tag, whereas the C-terminal GFP fusion did not change the localization [110,111]. Patch clamp analysis of GFP-AQP6-expressing HEK293 cells revealed a previously unknown anion transport mechanism with high nitrate permeability for AQP6.

These results underline that confocal microscopy studies using GFP fusion proteins do not always reveal the subcellular localization of the native protein, and therefore strongly suggest using both, N- as well as C-terminal fluorophore attachments as a common standard for tag-based localization experiments. Furthermore, surface expression of intracellular transport proteins upon manipulation of sorting/internalization signals represents a powerful tool for their functional characterization, which in some cases even was the only successful approach so far. There is also one report, in which the use of inhibitors of dynamin redirected the lysosomal K^+^ channel c to the plasma membrane upon expression in Xenopus oocytes [112].

Nevertheless, the successful expression in a different subcellular compartment does not guarantee the preservation of all properties of the wild type protein in its native membrane. Especially, lack of cosubunits, interacting proteins or regulatory factors, as well as differences in the lipid composition between the plasma membrane and the specific endomembrane might strongly affect the transport properties. It might therefore be beneficial or necessary to compare the properties of redirected proteins in several different cell types or membrane compartments like the plasma and vacuolar membrane (see Section 3.3). It should also be kept in mind that although an equivalent sorting motif may be present in two very similar transporters or channels, its mutation may not always have the same effect, as shown for the mammalian endosomal TPC1 versus the lysosomal TPC2 [95]. In contrast to TPC2, TPC1 did not reach the plasma membrane after eliminating one or both of its dileucine-based motifs.

In summary, several channels from the acidic lysosomal compartment of mammalian and plant cells were effectively redirected to the cell surface by mutation or masking of their lysosomal sorting/internalization signal (Table 3), which in most cases belong to the dileucin-based motives. Progress in understanding membrane protein trafficking in the different kingdoms will allow including further compartments of the secretory and endocytic pathway, such as the Golgi apparatus or early, late and recycling endosomes in future membrane transport research.

### 3.2. Nuclear Membrane Electrophysiology

The nuclear membrane may represent a novel heterologous system to express endo-lysosomal channels and transporters, as it provides relatively easy access to both the cytoplasmic and luminal sides of the membrane, so that ionic and ligand conditions can be rigorously controlled [113,114]. The technique, whose advantages and disadvantages have been summarized in Table 2, has been recently used to characterize the hTPC2 channel [115]. In their study, the authors generated a stable DT40TKO cell line expressing hTPC2 and lacking both functional InsP_3_R and RyR, two intracellular Ca^2+^ channels. Using the nuclear membrane patch-clamp technique, they detected a ~220 pS single-channel current activated by NAADP with K^+^ as the permeant ion.

### 3.3. Patch-Clamp Electrophysiology on Plant Vacuoles

#### 3.3.1. Sorting Routes and Signals to the Tonoplast and the Lysosomal Membrane

While performing more functions than the animal lysosome, i.e., storage of ions/metabolites and regulation of the turgor of the plant cell, the central lytic vacuole can be considered the lysosomal counterpart. This compartment (up to 40 μm in diameter) may occupy more than 80% of the cellular volume [116,117] in many cell types. Vacuoles are highly suitable for patch-clamp studies [118,119,120,121], because of simplicity of isolation and large size. For these reasons, many types of endogenous ion channels and transporters have been identified and characterized in great detail [116,122]. Besides macroscopic currents recorded in the whole-vacuole configuration [123,124], single channel events could be detected in excised vacuolar membrane patches, both in the cytosolic-side-out [125] and vacuolar-side-out configurations [126], Moreover, fluorescent indicator dyes have also been employed together with the patch-clamp technique [127,128,129,130,131]. The choice of ionic conditions and/or the use of appropriate Arabidopsis knock-out lines, allow to significantly reduce the density of the background currents, despite the presence of different types of endogenous ion transport systems.

Vacuoles can be used to study endolysosomal proteins only if these are sorted to the tonoplast. The finding that the main families of channels and transporters of the LM are delivered to the tonoplast in plant cells [26,132,133,134] provides exciting information and poses interesting questions on the evolutionary conservation of sorting mechanisms to the inner hydrolytic compartments.

From a topographic point of view, vacuoles and lysosomes occupy the same position within the secretory pathway: In general terms, they are one of the two endpoints of secretory traffic, the other being the cell surface. Like other secretory proteins, proteins of the tonoplast and the LM start their life in the ER [135]. The best characterized pathways to their final destinations pass through the Golgi apparatus and the trans-Golgi network (TGN). From there, a direct route proceeds via multivesicular bodies (MVB, also termed late endosomes) to the plant tonoplast or the LM, whereas an indirect route first leads to the plasma membrane and then to the final destination via endocytosis and MVB [136,137,138] (see Figure 5). Endocytosis of plasma membrane receptors and transporters for recycling or degradation in vacuoles and lysosomes operates in all eukaryotes, however to date the biosynthetic indirect route to the membrane of inner hydrolytic compartments has been described for lysosomes [136,139] but not for vacuoles. Therefore, it is not yet known whether missorting to the plasma membrane, observed for a number of tonoplast proteins upon mutagenesis or domain exchange with plasma membrane isoforms of the same gene family, is the result of actual direct missorting from the TGN to the cell surface or reflects lack of endocytosis from the plasma membrane [92,140,141,142].

Alternative ramifications can bypass some of the intermediate compartments. In plants, correct sorting of a number of tonoplast proteins is not blocked by brefeldin A or by dominant-negative mutant versions of Rab11 GTPases [143,144]. Brefeldin A causes fusion of ER and Golgi cisternae, inhibiting further anterograde Golgi-mediated traffic, whereas Rab11 GTPases regulate vesicle traffic at the TGN. Therefore, part of the tonoplast proteome can bypass the Golgi/TGN system, [137] and Figure 5. Indeed, there is growing evidence that the tonoplast of newly formed vacuoles in meristematic plant cells originates directly from the ER and is then expanded through Golgi/TGN/MVB-mediated traffic [145,146].

Membrane proteins are sorted along the endomembrane system through interactions between sorting motifs in their cytosolic domains and components of the different coat complexes that allow membrane traffic from one compartment to another. As introduced in Section 3.1, two classes of motifs present in the cytosolic head or tail of membrane proteins have been clearly identified as lysosomal and tonoplast sorting signals: Dileucine-based [D/E]XXXL[L/I] or DXXLL, and tyrosine-based YXXØ, where X can be any amino acid and Ø represents a bulky hydrophobic residue [136,137]. These motifs interact with components of the AP1-4 or GGA adaptor complexes necessary to recruit clathrin or other membrane coats that promote the formation of coated vesicles, either from the TGN, the plasma membrane or between early and late endosomes (notice that, in plant cells, the TGN also play the role of animal early endosomes; see Figure 5). Dileucine-based sorting signals for the tonoplast are recognized by AP1 [147], AP3 [142] or AP4 [148]. AP5 is involved in maintaining lysosome integrity [149], but its extact role has to be elucidate both in mammals and plants.

It should also be underlined that most membrane proteins also present ER-exit motifs that allow efficient initiation of traffic from this compartment, independently of their final destination. These diacidic (D/E-X-D/E), dihydrophobic or diaromatic (FF, YY, LL or FY) motifs interact with components of the COPII complex that initiates traffic from the ER (Barlowe, 2005; Marti et al., 2010). Finally, the ER quality control machinery usually prevents misfolded or unassembled newly synthesized polypeptides from trafficking [137]. Consistently with all these requirements for traffic and sorting, deletions or domain substitutions that destroy the tonoplast sorting signals often lead to mislocalization to the plasma membrane [94,97,141,142], whereas those that affect the ER-exit motifs or general folding result in ER retention [97,141,150].

The Golgi/TGN-mediated routes to the tonoplast and lysosomal membrane seem to use conserved mechanisms and signals [1]. This occurs despite the marked architectural differences in key compartments of the secretory pathway, perhaps most strikingly the Golgi apparatus: It is a single perinuclear membrane complex controlled by microtubules in most animal cells as opposed to a system composed of hundreds of apparently independent Golgi units moving along actin filaments all over any plant cell analyzed. These features rise as yet unresolved questions about the conservation of ER-to-Golgi and Golgi-to-endosome traffic mechanisms [151,152] and may also be related to the origin of the Golgi-bypassing routes to the tonoplast mentioned above. It has also been determined that potential N-glycosylation sequons are much less frequent in tonoplast proteins than in those of the plant plasma membrane, and that Golgi-modified Asn-linked oligosaccharide chains, abundantly present in plant plasma membrane proteins, are not detactable in tonoplast proteins of the same cells [153]. The major lysosomal membrane proteins are instead extensively N-glycosylated with oligosaccharide chains modified by Golgi enzymes. This “glycocalyx” is believed to protect the luminal loops of membrane proteins from degradation by lysosomal proteases [154]. In silico analysis of proteomes suggests that protection from vacuolar proteases has instead evolved by limiting the length of luminal domains of tonoplast proteins [153]. It is finally evident that a single large vacuole that occupies most of the cellular space in many fully expanded plant cells is quite different from the myriad of small lysosomes, at least in terms of surface/volume ratio.

#### 3.3.2. The Plant Vacuole as a Heterologous Expression System of Lysosomal Channels and Transporters

In line with similarity discussed above in trafficking and targeting between tonoplast and lysosomal membrane proteins, mutants plants from *Arabidopsis thaliana* lacking specific endogenous vacuolar channels or transporters can be used for the expression of the respective homologous animal lysosomal proteins (interestingly, oocytes of *Xenopus laevis* are the system of choice for ion channels and transporters localized in the plasma membrane [155,156,157,158,159]). Plants of Arabidopsis can be grown on soil in a growth chamber under controlled light and temperature conditions. The cDNA of the animal intracellular channel or transporter is cloned into a suitable plant expression vector conveying high protein expression. Fusion with a fluorescent marker may be helpful to verify the expression and localization of the protein. By using a well-established protocol [160], protoplasts can be transiently transformed; see Figure 6 for a schematic overview. The efficiency of transformation can be estimated by GFP fluorescence of the protoplasts. After one to four days, vacuoles can be easily released from transformed protoplasts for subsequent patch-clamp experiments.

The patch-clamp technique, developed by Neher and Sakmann [161], consists of establishing a tight contact (seal) between a glass pipette with a micrometric tip and the membrane of a cell, an organelle or an intracellular compartment (a vacuole in this case). When a seal is obtained, the resistance can reach values of several billion ohms with a consequent reduction of background noise [162] and the possibility of recording signals in the order of femtoamperes [163]. Patch-clamp measurements on plant vacuoles can be done in these four main configurations [164,165]: (i) Vacuole attached, which allows to measure currents across the membrane portion directly underneath the pipette tip, but with the limit of having no control over the content of the vacuolar lumen; (ii) whole vacuole, which allows macroscopic current recordings mediated by channels or transporters present on the entire vacuolar membrane; (iii) cytosolic-side-out and (iv) vacuolar-side-out excised patches, which are useful to detect single channel currents. A major advantage of vacuolar patch-clamp is that channel modulation by cytosolic factors can easily be studied, since the cytosolic side faces the bath solution which can be exchanged ad libitum during the experiment with a dedicated perfusion system, plant vacuoles (and protoplasts) are not firmly attached to the recording chamber as animal cells [134,166,167,168,169].

By applying the patch-clamp technique in whole vacuole configuration on isolated vacuoles from Arabidopsis mesophyll protoplasts lacking the endogenous AtCLCa, Costa et al. [127] recorded chloride–proton exchange activity of lysosomal CLC-7 from rat, which suggested the existence of an alternative CLC-7 operating mode, when the protein is not in complex with its auxiliary subunit OSTM1. In a similar approach, Boccaccio et al. [26] studied human TPC2 channels in AtTPC1 null background plants (Arabidopsis has a single gene whose protein, AtTPC1, is modulated by several factors [170,171,172]). They investigated TPC current responses to NAADP and PI(3,5)P_2_, underscoring the fundamental differences in the mode of current activation and ion selectivity between animal and plant TPC proteins and corroborating the PI(3,5)P_2_-mediated activation (see also [84]) and Na^+^ selectivity of mammalian TPC2. The endo-lysosomal hTPC1 channel was also sorted to the tonoplast, and its depencence on cytosolic and luminal calcium concentration could be fully characterized and mathematically modeled [133]. 

### 3.4. Patch-Clamp Electrophysiology on Giant Vacuoles from Yeast Cells

*Escherichia coli* and budding yeast (*Saccharomyces cerevisiae*) are frequently used to heterologously express plasma membrane proteins as complement assay to analyze their transport activity, similarly to other commonly used heterologous expression systems, such as *Xenopus laevis* oocytes and mammalian cell cultures. Budding yeast, a unicellular eukaryote, has a vacuole similar to plant cells. Its cell size is too small for whole-cell or whole- vacuolar patch-clamp measurements. Nevertheless, in 1990, vacuoles prepared from a tetraploid yeast strain, which are larger than haploid yeast cells, were employed and the vacuolar cation channel YVC1 was characterized [173]. As an alternative method for patch-clamp experiments on microorganisms, Yabe and co-workers developed a method to generate “giant” *E. coli* protoplasts (spheroplasts), as large as 5 to 10 μm in diameter, by digesting the cell wall by enzymatic treatment and addition of a peptidoglycan synthase inhibitor in the following incubation (named Spheroplast Incubation or SI method) [174]. Giant *E. coli* was used for patch-clamp recordings of H^+^ pump activity of respiratory chain F_0_ F_1_-ATPase [169]. Furthermore, the SI method was modified for yeast giant cell preparation using a 1,3-β-d-glucan synthase inhibitor. Enlarged yeast contained a huge central vacuole which apparently occupied more than 80% of the cellular volume. Using this system, the activity of yeast V-type H^+^-ATPase was evaluated directly from the vacuolar membrane [175]. Interestingly, the elaborate methods for inactivation of individual genes encoding endogenous channels in yeast enable the vacuole membrane to convert into a suitable expression platform with low background activity, providing a high signal-to-noise ratio for precise characterization of ion channels. In fact, various plant vacuolar proteins have been characterized using a biochemical approach with yeast mutant vacuoles [176,177,178]. This includes the detailed characterization of mung bean (*Vigna radiata*) proton-pumping pyrophosphatase (H^+^-PPase) [179] and the vacuole-localized K^+^ channel NtTPK1 from tobacco (*Nicotiana tabacum* cv. SR1) [180,181].

Yeast contains an ancestor of transient receptor potential (TRP) channel, designated as TRPY1 (previously named YVC1). TRPY1 displayed a relatively large ion conductance in the vacuolar membrane. Patch-clamp recordings on yeast vacuolar membrane showed that TRPY1 activity was activated by cytosolic Ca^2+^ and reducing agents and vacuolar lumenal Zn^2+^, where they were suppressed by phosphatidylinositol 3-phosphate and vacuolar lumenal Ca^2+^ [182]. TRPY1/YVC1-mediated Ca^2+^-regulated cation currents could be completely eliminated in the ∆*trpy1*/*yvc1* strain [183], which confirmed the same feature of TRPY1/YVC1 reported by [184]. The patch-clamp recordings of NtTPK1, Figure 7, revealed high selectivity for K^+^ and strong activation by cytosolic acidic pH and cytosolic Ca^2+^, which causes an approximately two-fold increase of the current amplitude of non-treated NtTPK1. The structural properties and Ca^2+^ and phosphatidyl inositol modulation of fungal TRP homolog from *Gibberella zeae* as well as TRPY1/YVC1 have been characterized [182,185]. Importantly, the human lysosomal channel hTPC2 could be also functionally characterised in this system [26].

The small genome size and rapid growth of *E. coli* and *S. cerevisiae* have contributed to the rapid progress of molecular biology and genetics. Disruption and induction of foreign genes is readily achieved without difficulty to obtain experimental material quickly. Exploiting mutant vacuoles with reduced channel background for patch-clamp experiments will enable the detailed characterization of endolysosomal ion transporters in eukaryotic cells.

## 4. Outlook on Novel Techniques Complementing Direct Functional Studies

### 4.1. Cryo-Electron Microscopy

Cryo-electron microscopy (cryo-EM) has the ability to provide 3D structural information of biological molecules and assemblies by imaging non-crystalline specimens (single particles). Latest advances in detector technology and software algorithms have allowed the determination of biomolecular structures at near-atomic resolution [186,187].

TRPML isoforms were investigated by cryo-EM to give new insights on molecular mechanisms of channel function and regulation. Li et al. [188] studied the molecular mechanism of the dual Ca^2+^/pH regulation of TRPML1 and of MLIV pathogenesis, focusing on the linker between the first two transmembrane segments of TRPML1. In lysosomes and endosomes, this linker faces the lumen (known as the ‘luminal linker’): Missense mutations affecting three single amino acids cause MLIV. They solved the crystal structures of the luminal linker in three different pH conditions (corresponding to the pH in lysosomes, endosomes and in the extracellular milieu) to elucidate its role in the dual regulation of TRPML1 by luminal Ca^2+^ and pH, TRPML1 channel assembly and MLIV pathogenesis. Moreover, cryoelectron microscopy structures of human TRPML3 were detected in the closed, agonist-activated and low-pH-inhibited states [189] revealing a new mechanism by which luminal pH and other physiological modulators such as PIP_2_ regulate TRPML3.

Very recently, cryo-EM was able to determine the structure of mouse TPC1 and later of the human TPC2 channel (in apo, ligand-bound open and ligand-bound closed states) providing insights into the mechanism of PI(3,5)P_2_-regulated gating [100].

The structure of chicken CLC-7 and human CLC-7 in complex with OSTM1 was also recently solved by cryo-EM [190].

### 4.2. Molecular Dynamics Simulations

X-ray crystallography and more recently cryo-EM provide us with an ever increasing number of atomic-resolution structures of membrane channels and transporters, allowing molecular dynamics (MD) simulations to study the mechanisms underlying their behavior [96,191,192]. Moreover, the increasing computational power permits simulations reaching tens or hundreds of microseconds, which are time scales approaching those of electrophysiological measurements.

Kirsch and colleagues [96], using a combination of MD simulations and functional analysis of TPC2 channel mutants, showed that PI(3,5)P_2_ cross-links two parts of the channel, enabling their coordinated movement during channel gating. In the last three years, several papers directly tackled the question of TPC’s function by employing molecular docking [34] and standard MD simulations combined with extended sampling methods such as metadynamics [193,194]. In the first paper, it was shown that TPC2 selectively binds naringenin, a molecule that modulates the hTPC2 activity in human cells [14,33]. By employing extended methods for sampling, the following two publications showed that the binding of PI(3,5)P_2_ activates mTPC1 and hTPC2 by increasing fluctuations in the hydrophobic gate (HG) region, and that sodium undergoes partial de-hydration when diffusing through this region. Finally, it was found that the occupancy of the central cavity (CAV) region is key for an effective sodium transport in hTPC2. The last result clearly suggests that at physiological concentrations optimum conductance is achieved when hTPC2’s CAV has a single occupancy.

Regarding transporters, the V-ATPase proton pump has been recently simulated [195] pointing out that the cooperative rearrangements of the three catalytic pairs are crucial for the rotation of the central stalk. In particular, V-ATPase interaction with lactoferrin was studied where it was shown that lactoferring inhibits V-ATPase by binding to the catalytic site [196]. The latter is particularly important as it is a mean of cytosolic acidification of fungi and cancer cells which leads to their subsequent death.

### 4.3. Genome Editing

Genome editing (also called gene editing) is a group of technologies that give scientists the ability to change an organism’s DNA. These technologies allow genetic material to be added, removed or altered at particular locations in the genome. Several approaches to genome editing have been developed. A recent one, known as CRISPR, has generated a lot of excitement in the scientific community because it is faster, cheaper, more accurate and more efficient than other existing genome editing methods [197]. CRISPR techniques allow scientists to modify specific genes while sparing all others, thus clarifying the association between a given gene and its role in the organism. One important application of this technology is to facilitate the creation of animal models with precise genetic changes to study the progress and possible treatment of human diseases. Recently, the involvement of localized Ca^2+^ release via TPC2 lysosomal channels was investigated using knockout of TPC2 (via the generation of a *tpcn2* mutant line of zebrafish using CRISPR/Cas9 gene editing): A significant attenuation of slow (skeletal) muscle cell differentiation was observed, suggesting that TPC2 is involved in the maturation of muscle cells [198].

### 4.4. Nanoscopy

Stimulated emission depletion (STED) microscopy is one of the techniques that make up super-resolution microscopy. It creates super-resolution images by the selective deactivation of fluorophores, minimizing the area of illumination at the focal point and thus enhancing the achievable resolution for a given system [199] bypassing the diffraction limit of light microscopy to increase resolution. This technique was used to reveal the close physical relationship between clusters of ryanodine receptors (RyRs) in the terminal cisternae of the sarcoplasmic reticulum (SR) and TPC2 channels on the lysosomal membrane. It has been proposed that TPC2-RyR clusters act as “trigger zones” in which TPCs are stimulated to create highly localized elementary Ca^2+^ signals that subsequently lead to the opening of the RyR in the SR/ER membrane, resulting in global signals via Ca^2+^-induced Ca^2+^ release [198].

## 5. Conclusions

Lysosomal channels are emerging as fundamental proteins for cellular homeostasis and as a pharmacological target for a multitude of human pathologies. Here, we presented the main methods capable of revealing their functional properties. It is very likely that the advent of new technologies will further expand our knowledge on this type of protein in order to fully clarify their role in their metabolic context.

## Figures and Tables

**Figure 1 cells-11-00921-f001:**
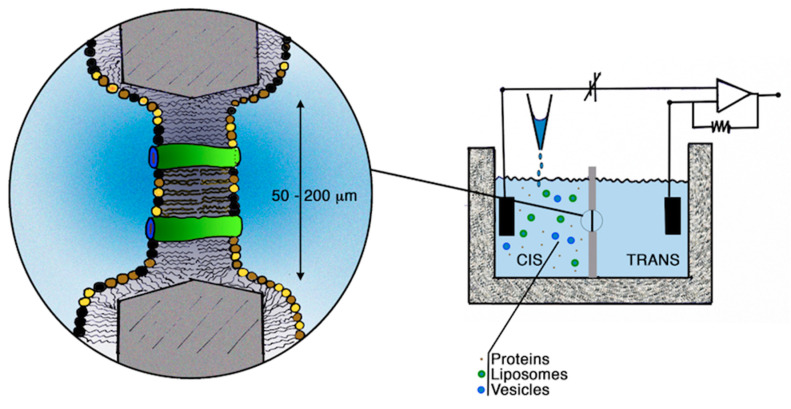
Schematic representation of a planar lipid bilayer system. Left: Solvent-free planar lipid membranes (PLM) (also called folded bilayers) are typically formed across a hole (~50–200 μm) present in a thin Teflon septum dividing two distinct compartments drilled in a Teflon chamber. The lipids are dissolved in n-alkane solvents (typically n-hexane) and spread at the air–water interface of the ionic solutions present in both compartments. The lipid bilayer is formed by the spontaneous apposition of the hydrocarbon chains of the monolayers when the liquid levels are simultaneously raised above the aperture as described by [56]. Alternatively, planar bilayers can be formed by dissolving the lipids in n-decane that can be painted or brushed onto a larger (~1 mm) hole to form so-called painted membranes or black lipid membranes (BLM) [57]. Unfortunately, there is some concern about the possibility that some n-decane molecules remain in the bilayer affecting the properties of the transport proteins. Improved techniques have recently been developed to obtain more stable and high-performance lipid bilayers to study channels/transporters under conditions as close to natural conditions as possible (for a review, see [58]). Right: Proteins can be directly added to the monolayer(s); however, this approach frequently destabilizes the formation of the folded bilayers; as an alternative they can be added to the bath solution of the cis (as well as the trans) compartment(s) pending the stochastic interaction with the bilayer and the subsequent incorporation. Furthermore, proteins can be incorporated into amphiphilic spherical systems (see Figure 2), such as liposomes or membrane vesicles (here shown not to scale) operating as protein carriers to the bilayer. The electrical properties of the protein enriched bilayer can be investigated by means of Ag/AgCl electrodes that connect the chamber compartments to the headstage of a patch-clamp amplifier. The ion transport properties of the channels/transporters result in variations of the very tiny current flowing through the hydrophobic lipid bilayer almost impermeable to the ionic fluxes. The currents are typically recorded under voltage-clamp conditions controlled by a PC: Single channel transitions or macroscopic currents are recorded depending on the number of channels/peptides incorporated into the lipid bilayer phase.

**Figure 2 cells-11-00921-f002:**
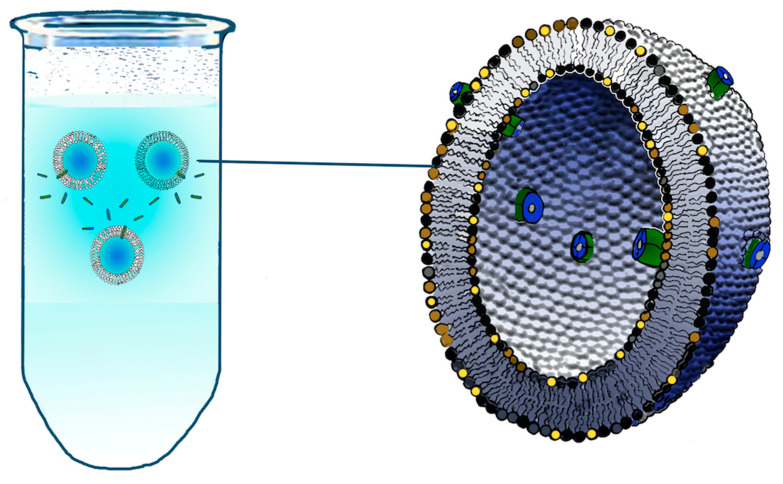
Schematic representation of liposomes containing channels/transporters. Liposomes are synthetic lipid vesicles where a lipid milieu separates the internal aqueous medium from the external ionic environment. After the dispersion of lipids in water solutions liposomes of different sizes (from 20–30 nm to several microns) and lamellarity can be obtained by changing the preparation method. Large and small unilamellar vesicles can be obtained from multilamellar vesicles by different methods such as ultrasonic treatment, or several extrusion cycles either through a small orifice under high pressure or through a polycarbonate membrane [59] (for a complete review, see Ref. [60]). Protein enriched liposomes (proteoliposomes) containing ion channels/transporters can be reconstructed from an incredible number of lipids that incorporate purified transport proteins from a variety of plasma or organellar membranes [60]. In turn, the reconstitution of cell/organellar planar membranes is achieved by the fusion of proteoliposomes to preformed PLM or BLM. Furthermore, native vesicles can be obtained by standard methods of cell fractionation, homogenization and centrifugation of native plasma as well as organellar membranes, including lysosomes [55,61]. Liposome and vesicle fusion with artificial bilayers can be facilitated by different procedures and tricks such as the presence of organic n-alkane solvents in the bilayer, the stirring of the bath solutions, the addition to the bath of millimolar concentrations of divalent cations, the existence of an osmotic pressure between the solutions present in the two compartments, the presence of nystatin and ergosterol in the proteoliposomes or even by centrifugal forces.

**Figure 3 cells-11-00921-f003:**
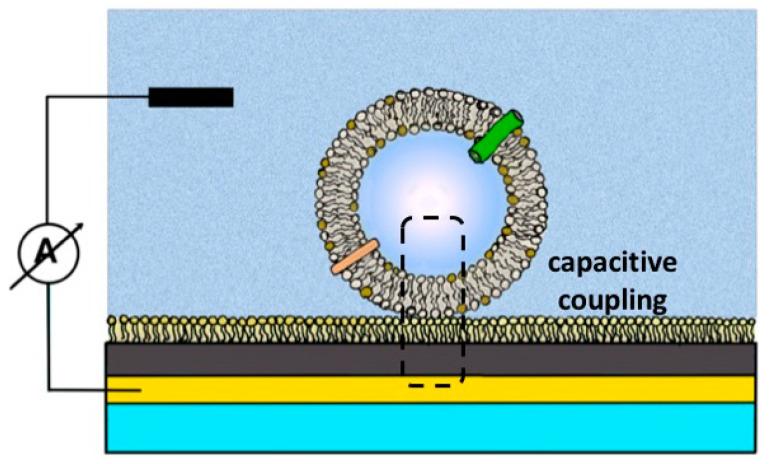
Schematic representation of the system for electrophysiological measurements on solid-supported membranes. SSM comprises a glass slide (light blue) covered by a very thin gold layer (in yellow) that binds thiol groups (in grey) of an alcanethiol reagent. A lipid monolayer completes the system. Proteoliposomes or native membrane vesicles as well as membrane fragments are adsorbed to the lipid monolayer. A change in the charge across the proteoliposome due to the activation of ion channels or transporters induces a correspondent change in the charge across the SSM, which is supplied by the amplifier (A). Therefore, as indicated by the box, a capacitive electrical coupling is established. The transient currents due to this coupling provide information on the properties of the channels/transporters (in green and orange) present in the adsorbed means (for a detailed and exhaustive description, see [68].

**Figure 4 cells-11-00921-f004:**
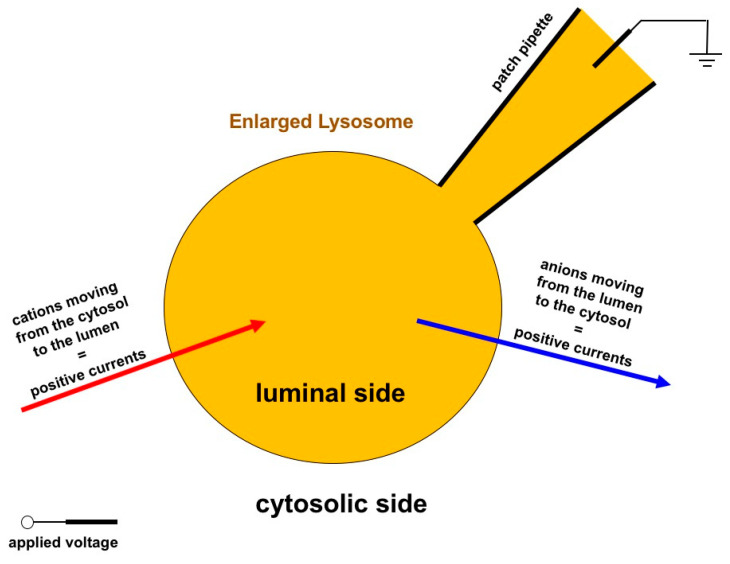
Cartoon of the patch-clamp recording configuration on enlarged lysosomes and current convention. The patch clamp technique is applied in the whole-lysosome configuration after lysosome enlargement by treatment with vacuolin-1. Positive currents correspond to the movement of cations from the cytosolic to the luminal side of the lysosome or to the opposite movement of anions.

**Figure 5 cells-11-00921-f005:**
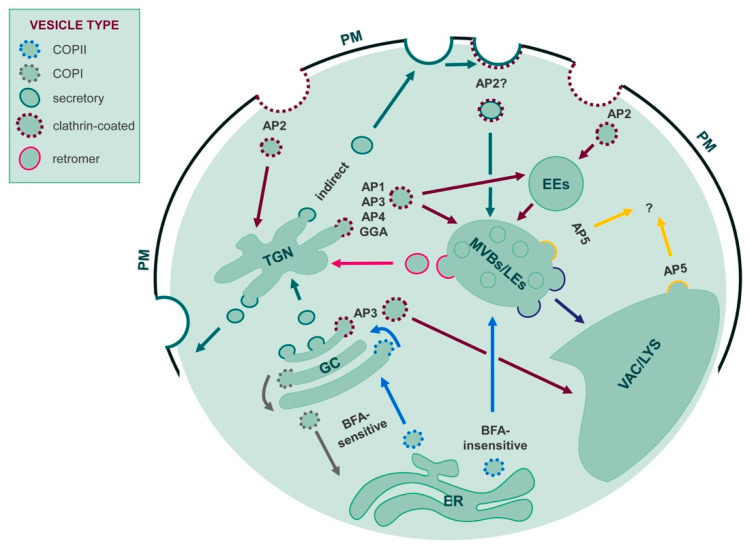
Biosynthetic and endocytic traffic routes to vacuoles/lysosomes. The color code of each arrow corresponds to that of the vesicle type involved. Traffic pathways are as follows. Blue, COPII mediated: BFA-sensitive ER to GC; anterograde intra GC cisternae; BFA insensitive ER to MVBs. Gray, COPI mediated: BFA sensitive GC to ER; retrograde intra GC cisternae. Wine, clathrin mediated: AP1-, AP3-, AP4-, GGA-mediated TGN to MVBs (plant cells) or TGN to EEs (animal cells); AP3-mediated GC to VAC (plant cells); AP2-mediated PM to TGN (plant cells) or EEs (animal cells). Notice that the TGN in plant cells also plays the role of animal EEs. Green, secretory vesicles: GC to TGN; TGN to PM. The TGN to MVBs/LEs indirect pathway could involve secretory vesicles from TGN to PM, followed by a possible AP2-mediated clathrin-coated vesicle internalization from PM to MVBs/LEs. Pink, retromer vesicles: MVBs/LE to TGN. Yellow: AP5-mediated traffic involved in maintaining lysosome integrity. Its exact role has to be elucidated, both in animals and plants. Violet: MVBs/LEs to VAC/LYS. Notice that this pathway seems to be a direct fusion event that does not require specific vesicles or coats. Abbreviations: ER, endoplasmic reticulum; GC, Golgi complex; TGN, trans-Golgi network; EEs, early endosomes; MVBs/LEs, multivesicular bodies/late endosomes; VAC/LYS, vacuole/lysosome; PM, plasma membrane; AP, adaptor protein complex; GGA, GGA adaptor-related proteins; BFA, Brefeldin A.

**Figure 6 cells-11-00921-f006:**
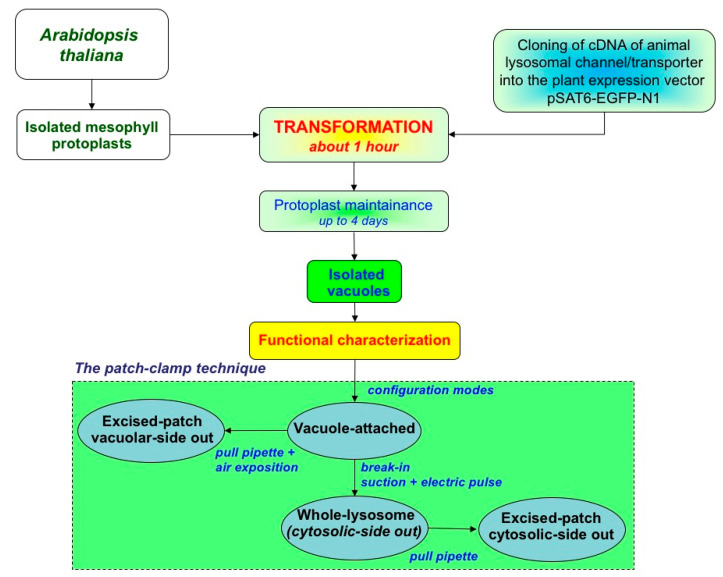
Flow chart of the experimental procedures showing how Arabidopsis vacuoles can be a heterologous expression system for animal lysosomal ion channels and transporters. Transformation can be performed on protoplasts from Arabidopsis wild-type plants or mutants lacking a specific endogenous channels or transporters.

**Figure 7 cells-11-00921-f007:**
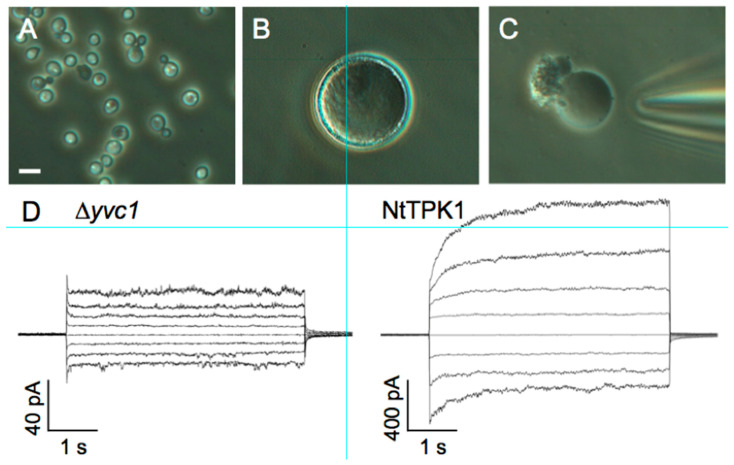
“Yeast Vacuoles”: The whole vacuole patch clamp recording of a giant yeast. (**A**–**C**) Haploid yeast (*S. cerevisiae*) was enlarged to giant yeast, and exposed vacuole with patch pipette. Bar = 5 µm. (**D**) Representative profile of whole vacuole currents elicited by a series of voltage steps ranging from −60 to 80 mV in 20 mV steps of the ∆yvc1 strain (left) and the NtTPK1-expressing cells (right).

**Table 1 cells-11-00921-t001:** Lysosomal channels and transporters (related references inserted in the main text).

Channel/Transporter	Transported Ion(s)
CLC-6	Cl^−^, H^+^
CLC-7	
SLC38A7	Na^+^, aminoacids
SLC38A9	
NHE3	Na^+^, H^+^
NHE5	
NHE6	
TPC1	Na^+^, Ca^2+^
TPC2	
VGCCs	Ca^2+^
TRPML1	Na^+^, Ca^2+^, Fe^2+^, Zn^2+^, cations
TRPML2	
TRPML3	
P2X4	
BK	K^+^
TMEM175	
LRRC8	Cl^−^, organic anions
V-ATPase	H^+^
CLN7	Cl^−^

**Table 2 cells-11-00921-t002:** Summary of current methods for the functional characterization of endolysosomal channels and transporters.

Method	Advantages	Disadvantages	Lysosomal Channels/Transporters
Incorporation into artificial membranes or liposomes	Low level of background current noise	Protein amount Channel removed from native environment Impurities	TPC1, TPC2, TRPLM1
Solid-supported membrane-based electrophysiology	Native environment Automation Suitable for screening	No control of membrane potential No control of luminal solution	CLC-7, V-ATPase
Flux measurements on purified lysosomes	Native environment Large number of lysosomes tested	No control of membrane potential No control of luminal solution	CLC-7
Patch-clamp electrophysiology on enlarged lysosomes	Native environment Direct Robust	Insufficient resolution to detect the activity of low turnover rate transporters Interference by endogenous channels and transporters Need of trained electrophysiologist	TPCs, TRPML1, BK, LRRC8, TMEM175, LRRC8, CLN7
Targeting to the plasma membrane upon manipulation of sorting signals	Well-characterized expression systems can be used Easy to perform, even if sorting/retention signals are not known	Different lipid environment may affect activity Modification of protein by mutation or tag Not applicable to all intracellular transmembrane proteins	CLC-6, CLC-7, TPC2, GLUT8, LRRC8, CLN7
Nuclear membrane electrophysiology	Easy access to cytoplasmic and luminal sides Ligand conditions rigorously controlled High temporal resolution Simple protocol High signal-to-noise ratio	Not tolerant to high V_app_ Low quality and stability of giga-ohm seals High background current Difficult excised nuclear patches	hTPC2
Patch-clamp electrophysiology on plant vacuoles	Good knowledge of vacuolar endogenous channels Large size Ease of isolation Low noise Possibility of different patch configurations Eukaryotic post-translational modifications	Long protoplasting procedure Need of trained electrophysiologist Fused fluorescent protein could impair functionality Differences with mammalian post translational modifications	CLC-7, hTPC1, hTPC2
Patch-clamp electrophysiology on giant vacuoles from yeast cells	Very high signal-to-noise ratio	Need of preparation of giant cells	hTPC2

**Table 3 cells-11-00921-t003:** Intracellular transport proteins functionally expressed in the plasma membrane after manipulation of sorting/internalization motifs (related references inserted in the main text).

Transport Protein	Name	Origin	Localization	Expression System
Monosaccharide facilitator/glucose transporter	GLUT8	*Mammalia*	Late endosomes, lysosomes	Xenopus oocytes
Cystine/proton symporter	Cystinosin	*Mammalia*	Lysosomes	Xenopus oocytes; COS cells
Aquaporin	AQP6	*Mammalia*	Acidic vesicles	HEK293 cells; Madin–Darby canine kidney cells
Sialic acid/proton symporter	Sialin	*Mammalia*	Late endosomes, lysosomes	HEK293 cells
Nucleoside transporter	ENT3	*Mammalia*	Late endosomes, lysosomes	Xenopus oocytes
Monosaccharide facilitator/glucose transporter	ESL1	Plant	Vacuole	Tobacco BY2 cells
Two-pore cation channel	TPC2	*Mammalia*	Late endosomes, lysosomes	HEK293 cells; Xenopus oocytes
Chloride/proton antiporter	CLC-6	*Mammalia*	Late endosomes	Xenopus oocytes; CHO cells
Chloride/proton antiporter	CLC-7	*Mammalia*	Lysosomes	Xenopus oocytes; HeLa cells
Cationic amino acid transporter	PQLC2	*Mammalia*	Lysosomes	Xenopus oocytes
Two-pore cation channel	TPC1	Arabidopsis	Vacuole	HEK293 cells

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
