# Peer review of "Current Methods to Unravel the Functional Properties of Lysosomal Ion Channels and Transporters"

_cells, 2022, doi:10.3390/cells11060921_

Round 1
Reviewer 1 Report
The manuscript summarizes the techniques used to study the activity of lysosomal channels and transporters, with an emphasis on proteins transporting inorganic ions rather than metabolites.
Unlike plasma membrane proteins, functional studies of lysosomal transport proteins are hindered by their intracellular location. Various approaches have been developed to overcome this technical barrier. The manuscript is a fair, comprehensive review of current methods.
Specific points:
- Fig 3 is poorly informative. The scheme should explain how proteoliposome polarization by ion channels/transporters induces a capacitative current in the gold layer.
- P. 10, lines 294/295 and 304/305: the measurements of equilibrium potentials in ref. 15 (using valinomycin to clamp the lysosome potential at diverse K+ equilibrium values) contradicts the statement that purified organelles "do not allow to control the membrane potential" (lines 304/5 and table 2). Authors should rephrase this word of caution more accurately.
- P. 13, last paragraph: functional studies of the lysosomal cystine transporter have been performed on a construct lacking only the classical sorting motif in the C-terminal tail. Deletion of the noncanonical motif (YFPQA peptide, probably not a real sorting motif) in fact abrogates the transport activity (ref. 94).
Author Response
Answers to Referee 1
Comments to the Author
“The manuscript summarizes the techniques used to study the activity of lysosomal channels and transporters, with an emphasis on proteins transporting inorganic ions rather than metabolites … The manuscript is a fair, comprehensive review of current methods.”
We thank the Referee for the positive evaluation of our manuscript.
1) “Fig 3 is poorly informative. The scheme should explain how proteoliposome polarization by ion channels/transporters induces a capacitative current in the gold layer.”
Following the Referee’s suggestion, we modified Fig. 3 inserting a box to highlight that the coupling between the proteoliposome and experimental support is not resistive but capacitive. We also added the following explanation in the figure legend: “A change in the charge across the proteoliposome due to the activation of ion channels or transporters induces a correspondent change in the charge across the SSM, which is supplied by the amplifier (A). Therefore, as indicated by the box, a capacitive electrical coupling is established.”
2) “P. 10, lines 294/295 and 304/305: the measurements of equilibrium potentials in ref. 15 (using valinomycin to clamp the lysosome potential at diverse K+ equilibrium values) contradicts the statement that purified organelles "do not allow to control the membrane potential" (lines 304/5 and table 2). Authors should rephrase this word of caution more accurately.”
In accordance with the referee's opinion we have modified the sentences as follows:
“Finally, the equilibrium potential for H+ flux, monitored by BCECF, was measured at a series of theoretical voltages set with K+/valinomycin.”
“It requires to perform radioactivity measurements and it does not allow direct and precise control of the membrane potential”
3) P. 13, last paragraph: functional studies of the lysosomal cystine transporter have been performed on a construct lacking only the classical sorting motif in the C-terminal tail. Deletion of the noncanonical motif (YFPQA peptide, probably not a real sorting motif) in fact abrogates the transport activity (ref. 94).
The referee was right: partial relocalization produced functional transporters, the cytosolic loop is required for activity. We corrected as follows (see lines 470-475 of the amended manuscript): “This was sufficient to study the proton-translocation for this unusual seven-transmembrane domain protein, which is distantly related to the proton pump bacteriorhodopsin [94,95]. A complete plasma membrane relocalization required the additional deletion of a novel, non-canonical lysosomal sorting motif (YFPQA) in a cytosolic loop [93], which in turn abrogated the transporter function [94].”
We thank this Referee for his help in improving the quality of our manuscript.

Reviewer 2 Report
This review by Festa et al., provides an overview on the techniques available for the functional characterization of lysosomal channels and transporters. More specifically the authors have tried to summarize the vast literature on the methods used to unravel the functional properties of lysosomal ion channels, transporters and in parallel have discussed their applications, advantages, disadvantages of the methods. Overall, the manuscript is convincing. However it is way too lengthy and therefore it becomes difficult to follow. The manuscript can be shortened to an large extent if the authors work on the language, style and I urge the authors to do so.
Further, I recommend the authors carry out the following minor corrections:
Page 1, Line 41-43: Wordy “In fact, lysosomes play a key role in maintaining cellular homeostasis.” (degradation, recycling, autophagy, cell death, cell proliferation, cell defence, immunity autoimmunity processes = maintenance of cellular homeostasis)
Page 2, Table 1: Please make a clear distinction by drawing a line underneath each channel/transporter (as done in Table 2) so that the ions transported by each of those channel/transporters remains clear
Page 2, Line 66: CLC - Please indicate what CLC stands for (there are couple of such gene, protein names in the manuscript (SLC, NHE,….) which requires expansion when introduced the first time).
Page 3, Line 126: The first part of the sentence appears incomplete. “A summary of the methods which will be reviewed in the following?…..”
Page 5, Table 4: For better reach, clarity column 4 (Lysosomal channels/ transporters) should be made as column 2
Page 6, Line 148: Figure 1. LegendsSchematic representation of a planar lipid bilayer system.
Page 6, Line 149-170: The figure legend (left & right) is way too long, difficult to follow. Please stick to describing the figure and be concise.
Page 7, Line 177-183: Similar comment as above. Please stick to describing the figure and be concise.
Page 8, Figure 3: Please use the white space next to the figure and label the figure directly instead of mentioning it by text in the legend (SSM comprises a glass slide (light blue) covered by a very thin gold layer (in yellow)). This will easily save few lines in the text and makes readers life easy as well.
Page 15, Section 3.2 Nuclear membrane electrophysiology: Out of context – The lines 486-495 can be removed.
Page 15, Section 3.3. Patch-clamp electrophysiology on plant vacuoles: The subsections underneath this section (3.3) is incredibly long/wordy and is difficult to follow. The authors must work on it and be more concise.
Page 18, Line 616: …underneath the pipette pip, but… typo (must be tip)
Page 21, Section 4.3. Genome editing: Line 742-756 - Does not add/contribute much to this review (wordy once again). Can be removed
Author Response
Answers to Referee: 2
Comments to the Author
This review by Festa et al., provides an overview on the techniques available for the functional characterization of lysosomal channels and transporters. … Overall, the manuscript is convincing.”
We thank the Referee for the positive evaluation of our manuscript.
1) “However it is way too lengthy and therefore it becomes difficult to follow. The manuscript can be shortened to an large extent.”
We comprehend the referee’s point. However, since the paper was written by many authors, we believe that shortening the manuscript would make the work losing completeness (see also our comment below).
2) Page 1, Line 41-43: Wordy “In fact, lysosomes play a key role in maintaining cellular homeostasis.” (degradation, recycling, autophagy, cell death, cell proliferation, cell defence, immunity autoimmunity processes = maintenance of cellular homeostasis)
According to the Referee’s suggestion, we modified the sentence as follows: “In fact, lysosomes represent the key players in degradation, recycling, autophagy, cell death, cell proliferation, cell defence, immunity-autoimmunity processes and therefore in maintenance of cellular homeostasis”.
3) Page 2, Table 1: Please make a clear distinction by drawing a line underneath each channel/transporter (as done in Table 2) so that the ions transported by each of those channel/transporters remains clear
The Referee was right, the table was not clear. We modified accordingly.
4) Page 2, Line 66: CLC - Please indicate what CLC stands for (there are couple of such gene, protein names in the manuscript (SLC, NHE,….) which requires expansion when introduced the first time).
Done.
5) Page 3, Line 126: The first part of the sentence appears incomplete. “A summary of the methods which will be reviewed in the following?…..”
We modified the sentence as follows: “A summary of the methods, their respective advantages and disadvantages, together with the lysosomal channels/transporters to which they have been applied, are presented in Table 2.”
6) Page 5, Table 4: For better reach, clarity column 4 (Lysosomal channels/ transporters) should be made as column 2
We modified table 2 on page 5 to improve its readability.
7) Page 6, Line 148: Figure 1. LegendsSchematic representation of a planar lipid bilayer system.
Done. We thank the Referee for this correction.
8) Page 6, Line 149-170: The figure legend (left & right) is way too long, difficult to follow. Please stick to describing the figure and be concise.
9) Page 7, Line 177-183: Similar comment as above. Please stick to describing the figure and be concise.
10) Page 8, Figure 3: Please use the white space next to the figure and label the figure directly instead of mentioning it by text in the legend (SSM comprises a glass slide (light blue) covered by a very thin gold layer (in yellow)). This will easily save few lines in the text and makes readers life easy as well.
11) Page 15, Section 3.2 Nuclear membrane electrophysiology: Out of context – The lines 486-495 can be removed.
12) Page 15, Section 3.3. Patch-clamp electrophysiology on plant vacuoles: The subsections underneath this section (3.3) is incredibly long/wordy and is difficult to follow. The authors must work on it and be more concise. 13) Page 21, Section 4.3. Genome editing: Line 742-756 - Does not add/contribute much to this review (wordy once again). Can be removed
We understand the Referee's point; however, as all of the above comments relate to the style of the manuscript, we kindly disagree in shortening or synthesizing the text.
This manuscript was written by many authors, each with their own sensitivity. We have made a great effort to make it homogeneous and we are satisfied with the result. We believe that the many details that we have deliberately pointed out may be of inspiration for the readers of Cells.
14) Page 18, Line 616: …underneath the pipette pip, but… typo (must be tip)
Tip is right; done. We thank the referee for this correction.
We thank this Referee for his careful reading and for helping to improve the quality of our manuscript.

Reviewer 3 Report
The manuscript by Festa et al. reviews the current methods available to test ion channel and transporter functionality. The review is well written, comprehensive and useful for the reader as it gives hints on the different methods to use, their advantages and disadvantages and their comparison. Therefore I recommend publications after addressing the following minor points:
-In Table 1, Table 2 and Table 3 a column with references should be added as it will be very useful to the reader to find immediately the references regarding a channel, a transporter or a method without going back to the text and looking for it.
-In line 140 the word lipid is repeated twice.
-In figure legend 1, line 150 the dimension of the hole seems to be 50-200 meters as there is only an m after 50-200.
-Figure 5. In mammalian cells endocytic vesicles coming from the PM reach first early endosomes (EEs) that are not included in this scheme and then there are MVBs/LEs. I think EEs should be included or at least there should be multiple arrows from PM to MVB/LE to indicate that there is something in between.
-Figure 5. The rappresentation of the ER is not correct as it seems to be the nucleus (with the nuclear envelope interrupted by the pores). ER membranes are in continuuum with the nuclear envelope but have a different morphology and they are not interrupted by pores.
Figure 5. The different pathways (indicated with arrows of different colors) should be defined in the figure legend. For instance: yellow arrows indicate the XX pathway, blu arrows indicate the XY pathway, etc.
Author Response
Answers to Referee: 3
Comments to the Author
“The manuscript by Festa et al. reviews the current methods available to test ion channel and transporter functionality. The review is well written, comprehensive and useful for the reader as it gives hints on the different methods to use, their advantages and disadvantages and their comparison.”
We thank the Referee for the positive evaluation of our study and manuscript.
1) In Table 1, Table 2 and Table 3 a column with references should be added as it will be very useful to the reader to find immediately the references regarding a channel, a transporter or a method without going back to the text and looking for it.
We thank the referee for the suggestion. Unfortunately, it is a matter of table and bibliography formatting our choice to insert the references in the main text. Moreover, in this way, tables are more readable.
2) In line 140 the word lipid is repeated twice.
Done. We thank the Referee for this correction.
3) In figure legend 1, line 150 the dimension of the hole seems to be 50-200 meters as there is only an m after 50-200.
Done. We thank the Referee for this correction.
4) Figure 5. In mammalian cells endocytic vesicles coming from the PM reach first early endosomes (EEs) that are not included in this scheme and then there are MVBs/LEs. I think EEs should be included or at least there should be multiple arrows from PM to MVB/LE to indicate that there is something in between.
5) Figure 5. The rappresentation of the ER is not correct as it seems to be the nucleus (with the nuclear envelope interrupted by the pores). ER membranes are in continuuum with the nuclear envelope but have a different morphology and they are not interrupted by pores.
6) Figure 5. The different pathways (indicated with arrows of different colors) should be defined in the figure legend. For instance: yellow arrows indicate the XX pathway, blu arrows indicate the XY pathway, etc.
Following the Referee’s comments we modified Figure 5 (see new Figure 5) and the related legend:
“The color code of each arrow corresponds to that of the vesicle type involved. Traffic pathways are as follows.
Blue, COPII-mediated: BFA-sensitive ER to GC; anterograde intra GC cisternae; BFA insensitive ER to MVBs.
Gray, COPI-mediated: BFA sensitive GC to ER; retrograde intra GC cisternae.
Wine, clathrin-mediated: AP1-, AP3-, AP4-, GGA-mediated TGN to MVBs (plant cells) or TGN to EEs (animal cells); AP3-mediated GC to VAC (plant cells); AP2-mediated PM to TGN (plant cells) or EEs (animal cells). Notice that the TGN in plant cells also plays the role of animal EEs.
Green, secretory vesicles: GC to TGN; TGN to PM. The TGN to MVBs/LEs indirect pathway could involve secretory vesicles from TGN to PM, followed by a possible AP2-mediated clath-rin-coated vesicle internalization from PM to MVBs/LEs.
Pink, retromer vesicles: MVBs/LE to TGN.
Yellow: AP5-mediated traffic involved in maintaining lysosome integrity. Its exact role has to be elucidated, both in animals and plants.
Violet: MVBs/LEs to VAC/LYS. Notice that this pathway seems to be a direct fusion event that does not require specific vesicles or coats.
Abbreviations: ER, endoplasmic reticulum; GC, Golgi complex; TGN, trans-Golgi network; EEs, early endosomes; MVBs/LEs, multivesicular bodies/late endosomes; VAC/LYS, vacu-ole/lysosome; PM, plasma membrane; AP, adaptor protein complex; GGA, GGA adaptor-related proteins; BFA, Brefeldin A.”
As the Referee can note:
1) We added the early endosomes and specified that in plants they are not present, because the TGN also performs the functions of the EEs.
2) we remade the endoplasmic reticulum
3) In the legend all the pathways are indicated
4) we added “(notice that, in plant cells, the TGN also play the role of animal early endosomes, see Figure 5).” at lines 627-628 of the main text.
We thank this Referee for his help in improving the quality of our manuscript.
